robotics/artificial intelligence

locomotion, modelling, reinforcement learning

**Author for correspondence:**
M. Calisti
e-mail: mcalisti@lincoln.ac.uk

# Learning to stop: a unifying principle for legged locomotion in varying environments

Thomas George Thuruthel[1], G. Picardi[2,3], F. Iida[1], C. Laschi[2,3,4] and M. Calisti[2,3,5]

[1]Bio-Inspired Robotics Laboratory, Department of Engineering, University of Cambridge, Cambridge, UK
[2]The BioRobotics Institute, Scuola Superiore Sant'Anna, Pisa, Italy
[3]Department of Excellence in Robotics and AI, Scuola Superiore Sant'Anna, Pisa, Italy
[4]Department of Mechanical Engineering, National University of Singapore, Singapore
[5]Lincoln Institute for Agri-food Technology, University of Lincoln, Lincoln, UK

GP, 0000-0001-9066-692X; CL, 0000-0001-5248-1043; MC, 0000-0002-2590-188X

Evolutionary studies have unequivocally proven the transition of living organisms from water to land. Consequently, it can be deduced that locomotion strategies must have evolved from one environment to the other. However, the mechanism by which this transition happened and its implications on bio-mechanical studies and robotics research have not been explored in detail. This paper presents a unifying control strategy for locomotion in varying environments based on the principle of 'learning to stop'. Using a common reinforcement learning framework, deep deterministic policy gradient, we show that our proposed learning strategy facilitates a fast and safe methodology for transferring learned controllers from the facile water environment to the harsh land environment. Our results not only propose a plausible mechanism for safe and quick transition of locomotion strategies from a water to land environment but also provide a novel alternative for safer and faster training of robots.

## 1. Introduction

Fundamental models of dynamic gaits (e.g. running, trotting, galloping, etc.) provide a high level picture of biomechanics and control of legged locomotion. Models such as the spring loaded inverted pendulum (SLIP, [1]) elucidate the fundamental relationships among legs' compliance, speed,

control, and shed a new light on the tight relationships among self-stabilizing mechanics and feedback control [2–5]. Besides its simplicity, the model successfully describes the behaviour of higher-order systems and it is employed as a reference for the control of single and multi-legged robotic devices [6,7].

Such generality also suggested the relevance of the SLIP model to locomotion in different media, i.e. in low gravity environments and in underwater environments. However, significant changes in the dynamics of the latter are difficult to explain by the SLIP model alone and, indeed, animals employ a slightly different gait in the underwater environment, which is referred to as *punting* [8]. An extension of the SLIP model, called underwater SLIP (USLIP, [9]), which takes into account the non-conservative nature of the system, the buoyancy, drag, and added mass effects of the punting gait, captures the dynamic of such locomotion [10], and it is successfully employed as a reference for the locomotion of underwater robots [11,12].

The similarity between the two models envision the possibility that an overall and unifying control strategy could be employed by animals and humans to move in both environments. In particular, it is demonstrated by evolutionary algorithms that locomotion learnt in water can be beneficial for land gaits [13], and similarly that smooth locomotion changes can be elicited by moving robots from land to water [14,15]. A similar conceptual hypothesis is proposed in this study, while we are looking for a control strategy which could be learnt and transferred from one environment to the other.

With this respect, reinforcement learning (RL) algorithms have been fairly successful in developing locomotion controllers. This is because of the complexity in modelling locomotion dynamics which makes learning-based approaches more suitable over analytical methods and the relative ease in defining the RL objective function [16,17]. In fact, the locomotion problem has become a popular benchmark task in the RL field. One of the earliest works on using RL for locomotion was done by Tedrake *et al.* [18]. The key aspect of their work was the design of a simple passive dynamic bipedal walker which greatly simplified the learning problem. RL for single-legged locomotion, a problem we are investigating in this work, was addressed using a model-free RL method called policy improvement using path integrals [19]. Their work showed how RL algorithms can be effectively used for exploiting the complex dynamics arising from compliant joints. Another interesting work showed how complex environment conditions can lead to the emergence of rich behaviours without explicit reward shaping [20]. Recent works have looked into deep RL algorithms for learning locomotion skills on physical prototypes [21,22].

A significant challenge in using RL algorithms for legged locomotion is the one of running real-world trials. Accurate simulation models are difficult to develop, especially with complex environmental interactions. Recent techniques like domain randomization can be a way around this problem, but it requires a large number of simulation trials and appropriate parameter settings [23]. Another solution, which this work will be adopting, is the concept of *shaping* [24,25]. *Shaping* is the idea of smoothly changing the physics of the problem to accelerate the learning process. The underlying principle is that by learning to solve problems in a simpler environment will facilitate the learning of a similar problem in a more complex version of the environment. Randlov proved that for a finite Markovian decision process with a limited reward signal, it is guaranteed that if a series of tasks converges to the original one, then the optimal value function converges to the original one as well [25]. In [26], a temporary device to reduce gravity helped the learning of single-leg hopping, showcasing the potential of shaping in real-world applications. The same concept is also known by the term curriculum learning in RL [27]. In this paper, we are proposing the transition from water to land, a landmark of earth colonization, as a naturally emerging *shaping* mechanism. We validate this hypothesis through simulations of fundamental legged models. For this, we propose the strategy of 'learning-to-stop' as a general locomotion plan that smoothly unifies a learned locomotion controller in both air and water. Using a deep RL framework, we show that the transfer of such learned controllers from water to air is faster and more efficient, as we would expect from an evolutionary viewpoint. We do not claim the animal water-land transition happened strictly following our approach, but our results confirm that a unifying model is possible and could be exploited for robust multi-domain locomotion.

The goals of this work are: to investigate the relationships between SLIP and USLIP, by finding a possible unifying strategy for the control of legged locomotion in multiple media; to establish the role of the environment on the learning progress of legged locomotion; and eventually to propose a learning procedure that could be employed in the training of effective legged devices, e.g. prosthetic devices or legged robots.

# 2. Material and methods

## 2.1. Background on previous fundamental models

The steady-state motion of the centre of mass (CoM) of animals performing various dynamic legged gaits can be described, abstracting the complexity of the animals' physical bodies [28], by a simple system consisting of a point-mass vaulting around a springy leg, the so-called SLIP ([1]). Extensive analysis of the SLIP model revealed the relationships between physical body properties (i.e. mass, leg length and leg stiffness) and locomotion features and stability [29] and stimulated research in the fields of biomechanics (e.g. [30,31]) and control (e.g. [32–34]). In robotics, SLIP gave great impulse to the development of hopping and running machines, which achieved unprecedented degrees of speed and agility through the integration of compliant elements carefully dimensioned to meet the criteria dictated by the model (e.g. [35–37]).

In SLIP, it is assumed that the resistance of air is negligible, thus the model is purely conservative and does not require the presence of an actuator to inject energy into the system. While this assumption is perfectly sensible in the case of terrestrial legged locomotion, the same cannot be said underwater, where the drag imposed on the body by a dense fluid introduces significant dissipation. Biological studies revealed fundamentals changes in the gait employed by crabs while running in water [8,10] that simply cannot be obtained through SLIP solutions. At the same time, there has been a growing interest towards underwater legged vehicles (ULR) [11,12,38–42], which can extend the capabilities of traditional underwater robots thanks to their improved interaction with the seabed. In order to extend the benefits of an approach based on the SLIP model to the underwater environment, an extension of SLIP which accounts for the contribution of water, namely USLIP, has been introduced [9] and employed as a reference model in the design and control of underwater legged machines [11,12,42]. The SLIP and USLIP schematics are reported in figure 1.

The equations of USLIP (equation (2.1)) are reported below:

$$
\begin{aligned}
\ddot{x} &= -\frac{X}{m+M}\dot{x}|\dot{x}| + \frac{k(x-x_t)}{m+M}\left(\frac{r_0 + r(t) - l}{l}\right) \\
\text{and} \quad \ddot{y} &= -\frac{Y}{m+M}\dot{y}|\dot{y}| + \frac{ky}{m+M}\left(\frac{r_0 + r(t) - l}{l}\right) - \frac{(\rho_w V - m)g}{m+M},
\end{aligned}
\tag{2.1}
$$

where $l = \sqrt{(x-x_t)^2 + y^2}$ is the length of the leg, $X = \frac{1}{2}c_x A_x \rho$ and $Y = \frac{1}{2}c_y A_y \rho$ are, respectively, the horizontal and vertical drag constants, $r(t) = r_s t$ is the linear leg elongation law and all variables and parameters are defined in table 1. Typically, the system is started in the *swimming phase* (foot not in contact with the ground and mathematically modelled by setting the leg stiffness $k = 0$) with initial conditions $[x_0, \dot{x}_0, y_0, \dot{y}_0]$ and touch down angle $\alpha$. In this phase, the agent follows a ballistic trajectory and dissipates its energy to drag until the *touch-down condition* ($y = l \sin \alpha$) is met. Here, the *punting phase* begins ($k \neq 0$), and the agent, following a minimal control law, extends its leg with constant velocity $r_s$ to actively compress the leg spring and gain elastic energy. A new swimming phase begins when the forces on the spring balance out as geometrically expressed by the *lift-off condition* $l = l_0 + r$. For adequate choices of the control parameters $\alpha$ and $r_s$, the system converges to period trajectories which correspond to stable locomotion patterns.

Up to now USLIP has only been used to model agents moving in water on planet Earth, thus setting $g = 9.81$ m s$^{-2}$ and $\rho = 1000$ kg m$^{-3}$. However, other environments can be modelled by the same equations by tuning the environment-dependent parameters of table 1. For example, running on land on Earth can be obtained with $g = 9.81$ m s$^{-2}$ and $\rho = 0$ kg m$^{-3}$ and running on the Moon with $g = 1.62$ m s$^{-2}$ and $\rho = 0$ kg m$^{-3}$. For these environments with negligible air density, the conservative SLIP model emerges by simply setting the leg extension speed $r_s = 0$.

## 2.2. Learning methodology

The common strategy for learning to locomote involves framing the problem as a RL problem with the objective to reach a specific velocity or maximize the locomotion speed while satisfying other user constraints like avoiding obstacles, reducing energy expenditure, etc. However, such learned policies will be highly specific to the particular objective and hence typically not generalizable and robust. Recent findings show that robustness can be achieved by learning in a rich environment [21] or using domain randomization techniques [23]. Yet, none of these techniques provide a methodology to

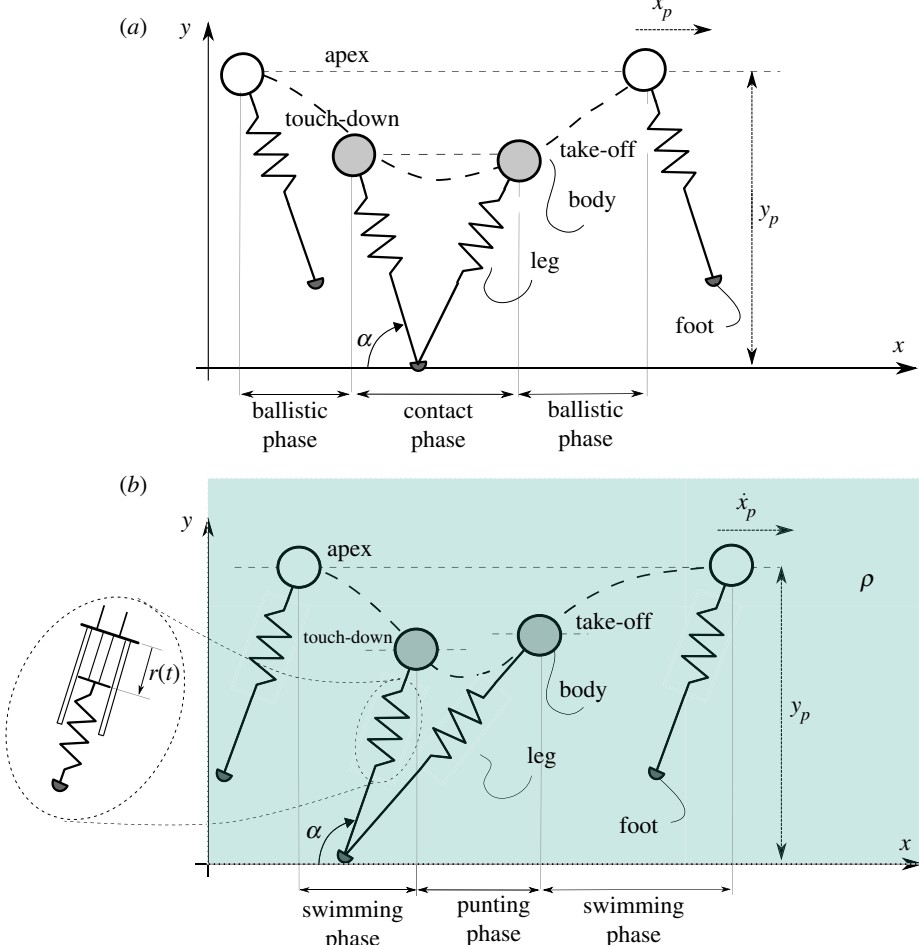

**Figure 1.** The SLIP (*a*) and USLIP (*b*) models with the state and control variables. The state of the system is full defined by the density of the environment $\rho$ and the horizontal velocity $\dot{x}_p$ and height of the model $y_p$ at the peak of each hop. The two control parameters are the angle of contact $\alpha$ and the elongation speed of the spring after contact $r(t)$.

develop a learned policy that can maintain stable locomotion while being able to control other state variables. In this work, we derive a strategy inspired from the RL approach used in autonomous flight, i.e learning to hover, or in our case, hop-in-place/learning-to-stop [43]. The idea is to learn a controller that brings the system to a zero horizontal velocity hop from the current state, in as few steps as possible. A closely related concept of capture point, where the strategy of placing the foot for stopping has been investigated for push recovery [44]. Hof showed that planning locomotion strategies based on the capture point locations, in contrast to the typical zero moment point formulations, allowed for stable walking in a simpler form [45]. Further studies showed the capture point methodology to be an effective and simple tool for the design of robust trajectory generators and feedback controllers for bipedal walking robots [46].

Once a controller for hopping-in-place is learned, tracking a set velocity can be done by offsetting the observed state of the model by the desired velocity to 'trick' the controller to move in the desired velocity. As the zero horizontal velocity is symmetric with respect to the design of the model, motion in either directions can be achieved without additional components. The next section presents the procedure for learning the hopping-in-place controller using a common RL algorithm and the control architecture of the horizontal velocity tracker.

### 2.2.1. Deep deterministic policy gradient

As mentioned before, our objective is to develop locomotion controllers capable of locomotion in different environmental conditions and investigate any underlying properties among the controllers. For developing the controller, we use a simple two-dimensional SLIP model that is placed in a

**Table 1.** All variables and parameters of the presented model. (Parameters are subdivided into control parameters, which can be modified online; design-dependent parameters, which relate to the intrinsic physical properties of the agent; and environment-dependent parameters, which model the interaction between the agent and the surrounding environment.)

| variables | |
|---|---|
| $x$ | CoM horizontal position |
| $\dot{x}$ | CoM horizontal velocity |
| $y$ | CoM vertical position |
| $\dot{y}$ | CoM vertical velocity |
| $x_t$ | horizontal foot position |
| **control parameters** | |
| $r_s$ | leg elongation speed |
| $\Delta r$ | maximum elongation |
| $\alpha$ | leg angle at touch down |
| **design-dependent parameters** | |
| $m$ | dry mass |
| $r_0$ | rest length of leg |
| $V_r$ | volume of the agent |
| $k$ | spring stiffness |
| $A_x$ | horizontal projection area |
| $A_y$ | vertical projection area |
| $\varsigma_x$ | horizontal drag coefficient |
| $\varsigma_y$ | vertical drag coefficient |
| **environment-dependent parameters** | |
| $g$ | gravity constant |
| $\rho$ | density of fluid |
| $M$ | added mass |

variable medium environment (figure 1). The state of the system is full defined by the density of the environment ($\rho$) and the horizontal velocity ($\dot{x}_p$) and height of the model ($y_p$) at the apex of each hop. The two control parameters are the angle of contact $\alpha$ and the elongation speed $r_s$ of the spring after contact. Note that the density of the environment can be directly estimated by the vertical acceleration at the peak, but for simplicity, we assume this information is directly available to the controller.

The deep deterministic policy gradient (DDPG) algorithm is a model-free, online, off-policy RL method [47]. A DDPG agent is an actor-critic RL agent that concurrently learns a value function and a policy. It uses off-policy data and the Bellman equation to learn the value function, and uses the value function to learn the policy (figure 2). The DDPG agent obtains the observation $S$ ($\dot{x}_p, y_p, \rho$) and reward from the model after each step and updates the actor and critic using a mini-batch of experiences randomly sampled from the experience buffer. Each episode can have a maximum of $n$ (5, in our case) steps and the model is reinitialized at a random state after the end of each episode. The reward obtained after each step is defined as:

$$\text{step reward, } R = \log\left(1 + 1/|\dot{x}_p|\right), \text{ if } y_p > \text{threshold, else}$$
$$R = 0 \tag{2.2}$$

As described in the previous section, the objective of the step reward is to reduce the horizontal velocity at the apex. A discount factor of 0.5 is multiplied to the reward value after each step to obtain the episode reward. This is done to favour faster convergence to the zero velocity state. The threshold for detecting failure is based on the height of the model at peak with respect to the ground. The action $A$ ($\alpha, r_s$) is chosen by the policy using a stochastic noise model at each training step. The

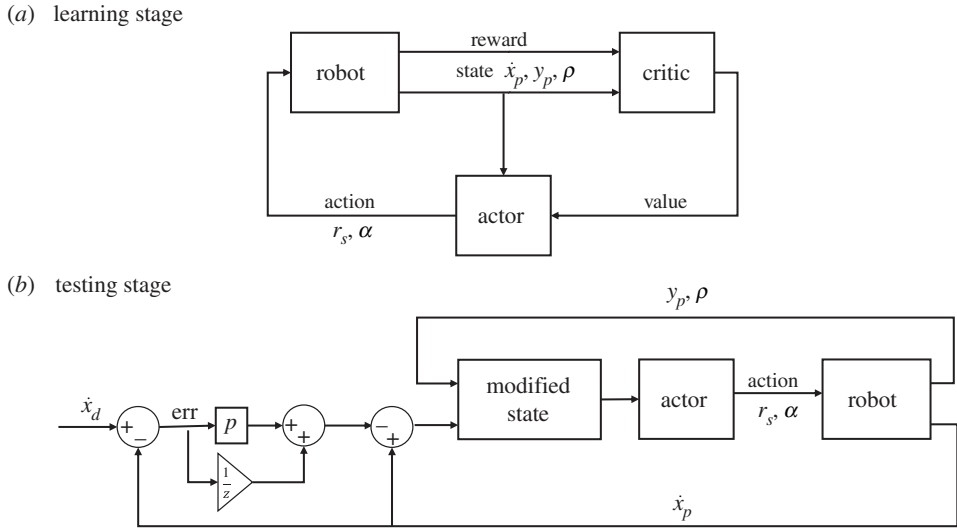

**Figure 2.** (*a*) The DDPG learning process. (*b*) The cascaded control architecture developed using the learned agent. Errors in velocity tracking are compensated by the proportional controller.

variance of the noise is gradually reduced by a constant factor during training to favour exploitation over exploration.

At the start of training, the DDPG algorithm creates the critic $Q(S, A)$ and the target critic $Q'(S, A)$ with the same random parameters. The critic, in our case, is a multi-layered neural network with rectified linear activation function (ReLu) activation layers. The output of the critic is the expected value for the given observation and action. The expected value is simply the sum of the current reward and the discounted future reward. Similarly, the actor $\mu(S)$ and the target actor $\mu'(S)$ is initialized with the same random parameters. The actor is also a multi-layered neural network with ReLu activation layers. At each step, the actor executes the action $A$, observes the reward $R$ and next observation $S'$. The experience ($S$, $A$, $R$, $S'$) is then stored in the experience buffer. For updating the parameters of the actor and critic, random mini-batches of experience from the replay buffer are used. A small replay buffer will cause the algorithm to overfit and create instability in learning, while a large replay buffer will slow down the learning process. The target networks are time-delayed copies of their original networks that slowly track the learned networks. Using these target value networks greatly improves stability in learning [47]. We use the MATLAB RL toolbox for initializing and training the networks. The source code of the algorithm and the trained agents can be downloaded from: https://github.com/tomraven1/DDPG-hop.

# 3. Results

This section presents the results of training an agent in various environments and how the strategy of transferring agents learned in a specific environment to another fares. Further analysis is done on the performance of the controller and the possibility of a universal controller for locomotion in water and air.

## 3.1. Training results

To test the learning algorithm in different mediums, two controllers are learned, separately in land and water. In order to introduce an understanding of medium properties on the controller strategy, the medium density is randomly initialized along with the other states for each episode. For the water environment, the medium density is defined as: $1 - |randn|/10$, (where 1 is the normalized water density), while for the land environment, the medium density is randomly initialized to: $0 + |rand|/10$. Here, randn is a standard normally distributed random number. The same network architecture and agent/critic parameters are used for both scenarios for comparison purposes.

The training progress, indicated by the average reward of the agent in both the mediums, is shown in figure 3. As observed in other works, learning to hop from scratch in a dense environment is much easier [9]. This is because of the low inertial effects in a dense medium combined with the stabilizing effect of buoyancy. When we transfer the pre-trained actor and critic from the water environment to the air

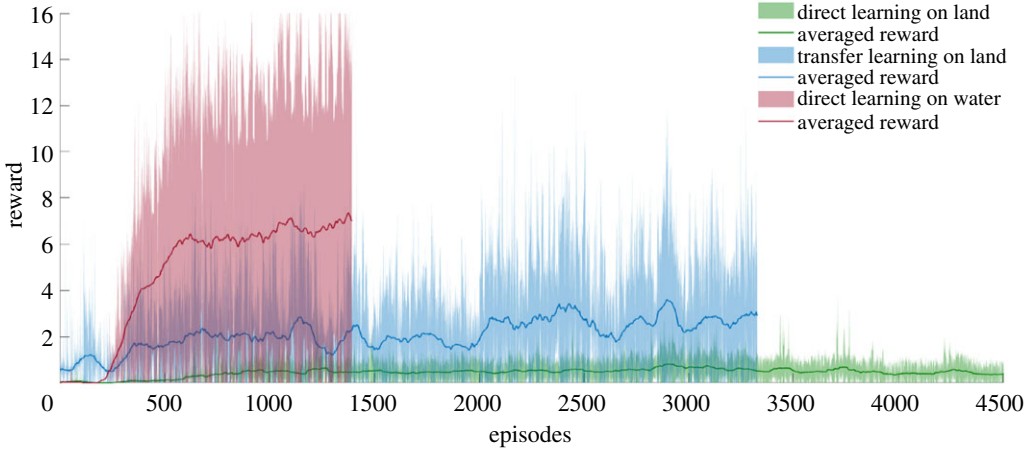

**Figure 3.** Agent reward during the training phase in different mediums. Transfer learning on land is done by retraining the actor and critic obtained in water.

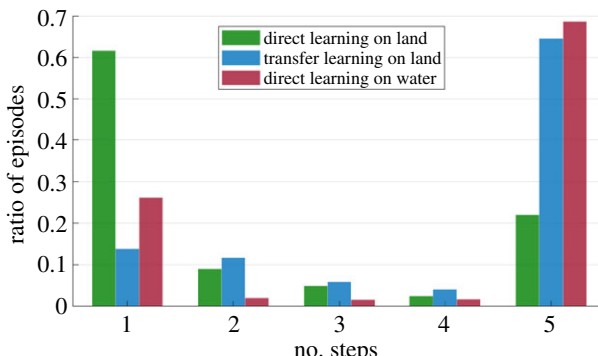

**Figure 4.** Number of steps during training for the first 1000 episodes shown as the ratio of the total episodes (a ratio of 1 implies that all the episodes resulted in the same number of steps). For our training, the maximum number of steps that the system can take for each episode has been set at 5. A lower number of steps most likely indicates a lower episode reward too.

environment, without the experience replay, we see that the agent converges to high reward much faster (figure 3). As the RL algorithm reaches a higher reward quickly in the water environment, even with the combined number of episodes, the transferred agent can obtain higher rewards than an agent trained directly on land. This implies that the actor and/or critic which learned in water and land must have significant similarities among themselves that is reflected in their parameter space too. Note that the variance and decay for exploration is also reinitialized for this network. So, the faster convergence is not because of a lower exploration parameter. As there is no additional cost in changing the medium (algorithmically), this is in fact a good strategy to guide your training, especially in cases where high rewards are sparse. Moreover, looking at the number of failures during training (figure 4), it is evident that a transferred agent has lower number of falls (steps <5). This is important for this case, as falling on land is more perilous than falling underwater. Hence, such a transfer strategy can be useful for safe training for real robots too.

The reason that learning to locomote in water is easier can be understood by looking at the reward landscape for stopping with respect to the control inputs for varying environmental mediums (figure 5). In the water environment, high reward regions are dense and continuous with respect to the control inputs. Hence, gradient-based approaches, like the one we adopt here, are well suited. On the other hand, for a pure air environment, the reward regions are sparse and discrete, making it difficult to be found. More importantly, when transitioning from water to land smoothly, we also observe a smooth variation in the reward landscape too, validating our hypothesis. This allows the agent which transferred from water to air to quickly adapt. Note that the reward function is almost independent from the elongation speed for the single-step case, but with multiple steps, the elongation speed makes an effect to the reward obtained by the agent. This is because a non-optimal elongation speed can drive the next state of the system to regions outside the bounds of the problem, making the

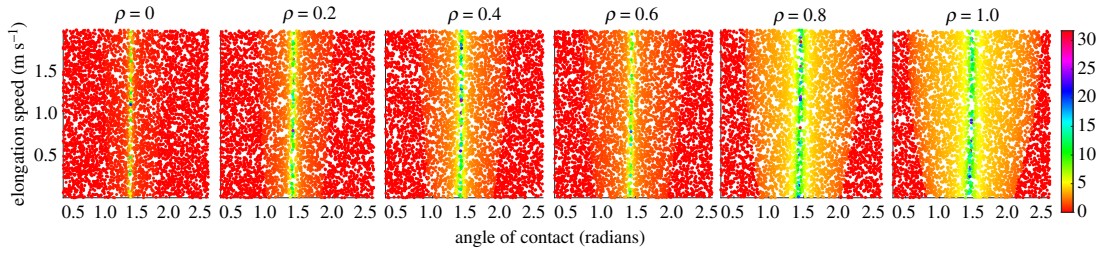

**Figure 5.** Reward landscape with respect to the control inputs for a fixed random initial state for different environmental mediums.

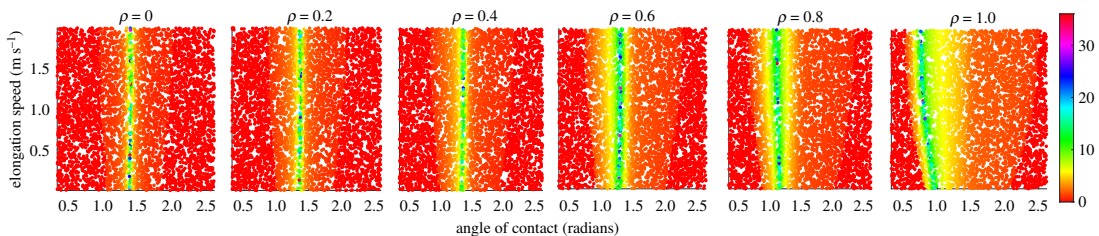

**Figure 6.** Reward landscape with respect to the control inputs for a fixed random initial state for different environmental mediums with a modified reward of moving at a fixed velocity of $1\,m\,s^{-1}$.

learning process unstable. For example, in the conservative air environment, a high elongation speed will eventually bring the height of the model to regions beyond the learning samples, causing instability in the learning process. In the water medium, a low elongation speed will eventually drive the model to a low height, causing the system to fall. Hence, to ensure optimal convergence of the elongation law, it is necessary to increase the maximum number of steps allowed for each episodes to a high value.

This smooth transition in the reward landscape is observed for all initial conditions. This is, however, only the case for the objective of minimizing the horizontal peak velocity. For other reward formulations, the transition across different mediums is not as smooth, although certain level of information can still be transferred from one medium to the other. One such example is shown in figure 6 for an objective to reach a horizontal velocity of $1\,m\,s^{-1}$. More flexible objectives like maximizing the horizontal velocity would have higher variations because of the diverse limits on the locomotion speeds in different mediums.

## 3.2. Controller analysis

This section presents the results of the learned controller. After the actor is trained, the agent can be directly used as the zero-velocity tracking controller. By offsetting the observed horizontal velocity of the model by the desired velocity and proportional integral derivative term of the error, we obtain a cascaded control architecture for controlling the locomotion speed while maintaining stability (figure 2). For simplicity, we only use a proportional component to stabilize the horizontal velocity. The base performance of the actor (zero desired velocity) is shown in figure 7. For all these analyses, we use an environment condition of pure water and air ($\rho = 1$ and $0$). The results of the controller show that the agent performs very well in water; reaching a stable limit cycle (i.e. hopping in place) in a few steps, with the horizontal velocity reaching almost zero. This is as expected, based on the training results. The agent trained directly on land, manages to take few steps, but is not stable enough to sustain locomotion. This explains the relatively low rewards obtained during training. The transferred agent is able to maintain stable locomotion and reaches a bi-periodic limit cycle (i.e. hopping around zero speed). The final velocities are near zero, but not as accurate as the case of the agent in water. This could be because of the lack of energy dissipating components in land, which makes it much harder to reach the low energy state of zero-velocity hopping. Looking at the control actions prescribed by the actor, we can see that the agent in water uses the maximum elongation speed for locomotion while the transferred agent uses the minimum elongation speed, essentially adapting to a passive SLIP system. A comparison of the solutions from the learned actor to the optimal solutions is shown in figure 8. The optimal solutions are obtained by performing a numerical optimization (pattern search) over the control space for one-step stopping. It can be seen that there are differences between the solutions from the agent and the optimal control solutions. This is because the RL agent can take

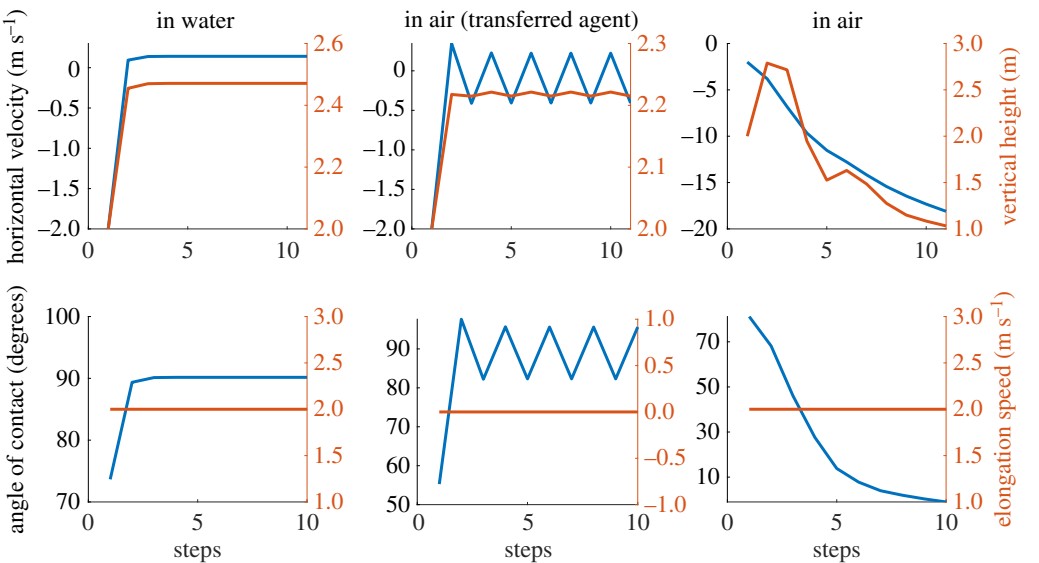

**Figure 7.** Base performance of the learned actor in different mediums. The agent is trained to minimize the horizontal velocity at apex at each step. The state/action values at the apex are only shown here.

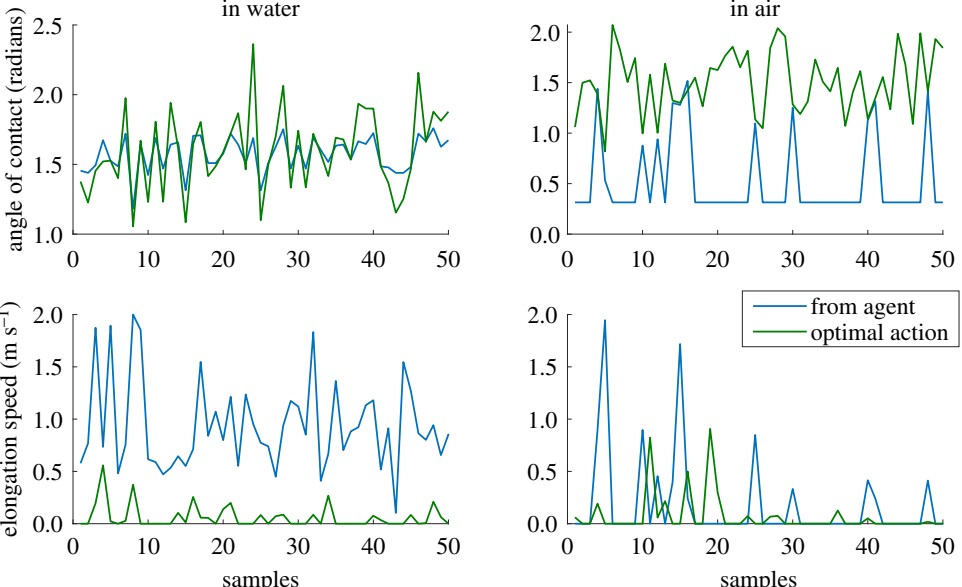

**Figure 8.** Solutions provided by the actor compared with the optimal solutions for one-step stopping.

more than one step to come to a stop. With longer training, we can expect the solutions of the controller to converge to the optimal solution as it would provide higher cumulative rewards.

The performance of the cascaded control architecture for the agent trained in water and transferred to land is shown in figures 9 and 10, respectively. As the agent which learned directly on land was not stable enough for longer periods of locomotion, we ignore that case here. For both cases, a desired velocity of 1 m s$^{-1}$ is prescribed and initialized from the same initial conditions. The tracked velocity for different $p$-values are shown in the results.

## 3.3. Robustness analysis

In this section, we investigate the robustness of the control policy to changes in the dynamical properties of the model. The robustness of the controller to noises in state estimation is already apparent as we rely on this robustness to control the horizontal velocity. The basin of attraction for the controller for changes in the mass of the body and the stiffness of the leg is shown in figure 11. This is obtained by initializing

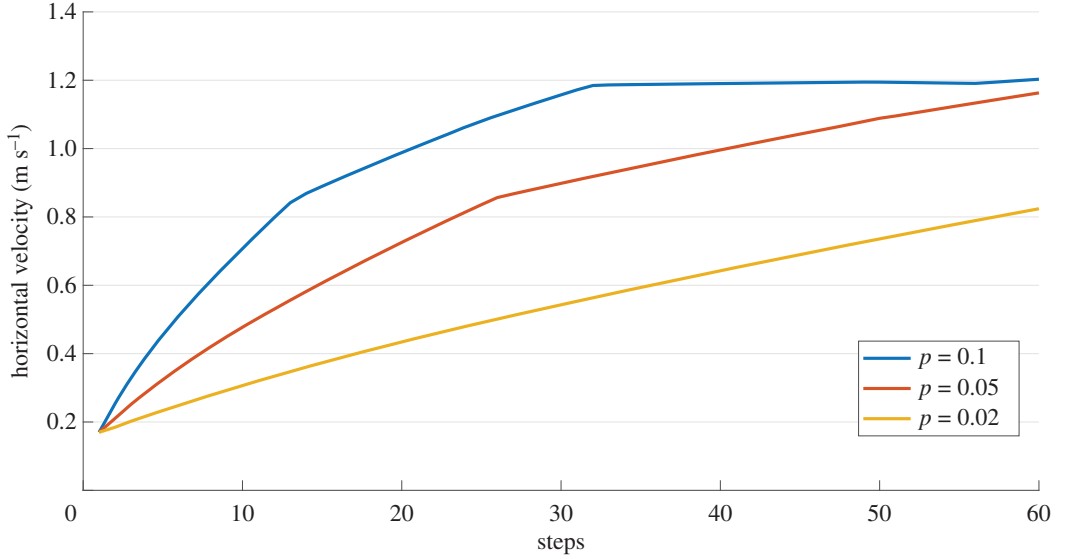

**Figure 9.** Performance of the cascaded controller in water for different $p$-values for a desired velocity ($\dot{x}_d$) of 1 m s$^{-1}$ (see figure 2 for reference). The horizontal velocity at peak is shown here.

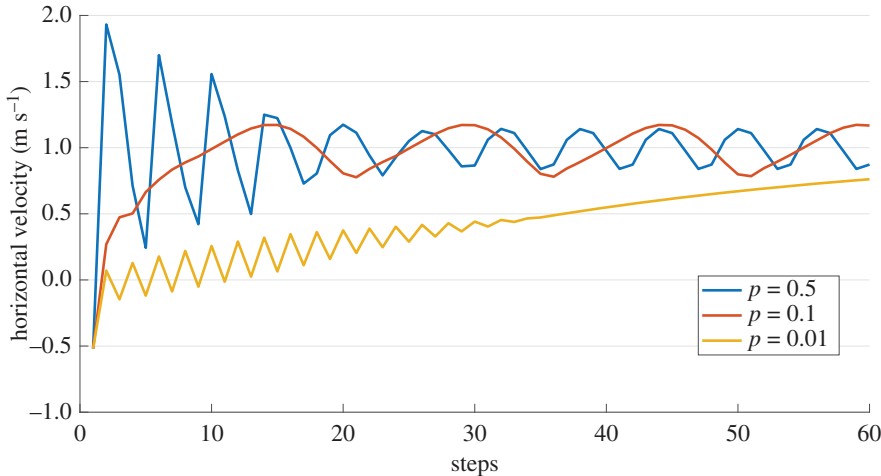

**Figure 10.** Performance of the cascaded controller in air for different $p$-values for a desired velocity ($\dot{x}_d$) of 1 m s$^{-1}$ (see figure 2 for reference). The horizontal velocity at peak is shown here.

the robot at random states and running the controller until the model takes 30 steps or it falls. We assume that the model is stable after 30 successful steps. The boundaries within which the control policy is stable defines the basin of attraction. As the dynamical properties are changed the basin of attraction would shrink as the control policy was learned for the default dynamical properties. However, it can be seen that even with drastic changes in the body parameters, the control policy is still stable for a large set in initial conditions. It can also be seen that the basin of attraction shrinks significantly as the stiffness decreases. Similarly, the maximum velocity achievable by the controller before falling is shown in figure 12. The maximum achievable velocity is a function of the dynamics of the robot, the actuator limits and our learning-to-stop strategy. If higher speeds are required, the learning objective has to be modified accordingly. However, this eliminates the generalizability of the controller as discussed before. The results however indicate that even with the current strategy the maximum speed achievable (6 m s$^{-1}$) is comparable to the average running speed of a human.

## 3.4. Gait analysis

In this section, we analyse the characteristics of an universal controller that is adapted to hop in both land and water. For this, we transfer the agent which learned in water to an environment that alternates

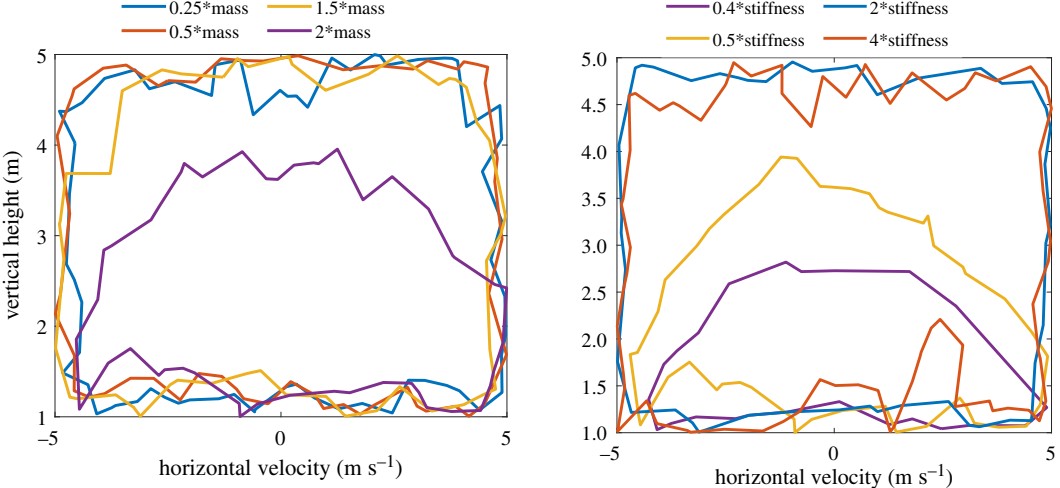

**Figure 11.** Basin of attraction for the transferred control policy in air for varying body dynamics.

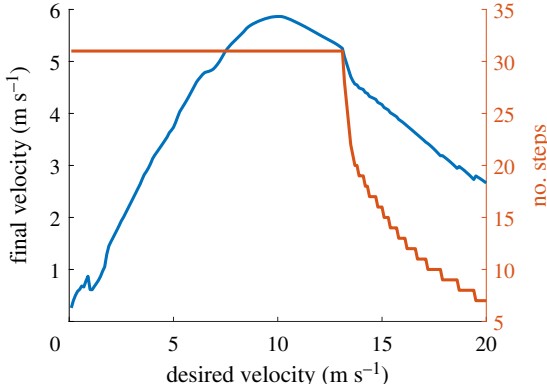

**Figure 12.** Maximum horizontal velocity achievable by the transferred control policy in air.

between pure water and air. The training is performed again with the same parameters as the previous sections. The maximum number of steps per episode is however increased to 10. The gaits observed for the single controller in both air and water is shown in figure 13. Here, we are showing the gait after convergence for a prescribed horizontal velocity of 1 m s$^{-1}$. Note that in air, the controller settles to the conservative SLIP model, characterized by symmetric touch-down and lift-off angles and in water, the controller settles to the USLIP model with acute touch-down and lift-off angles. As our model ignores many of the dissipative terms found in reality like friction and damping, the vertical displacements observed are higher than what is observed in nature.

## 4. Discussion

The seminal SLIP model was developed about 30 years ago, and since then several extensions were proposed, among the latter the USLIP model for underwater legged locomotion. Our results shed a light on the tight relationship between them, and on a possible unique fundamental model which represents legged locomotion on different media. As shown in figure 13, the learnt strategy is coherently employed in diverse media by adapting the angle of contact and elongation speed, the two key control parameters that differ between SLIP and USLIP, for the air and the water scenarios. This differentiates qualitatively the behaviours and it shows an energy-conservative behaviour for the air case, where energy is transferred to the elastic components of the leg during the deceleration of the CoM and further released as kinetic energy during the acceleration; and a energy-dissipated behaviour for the water case, where energy is dissipated by drag forces and it is injected by means of pushing actions. The model in our analysis being the same, the transition between the two diverse employments is dictated by the environment itself, through the reading of the density of the medium.

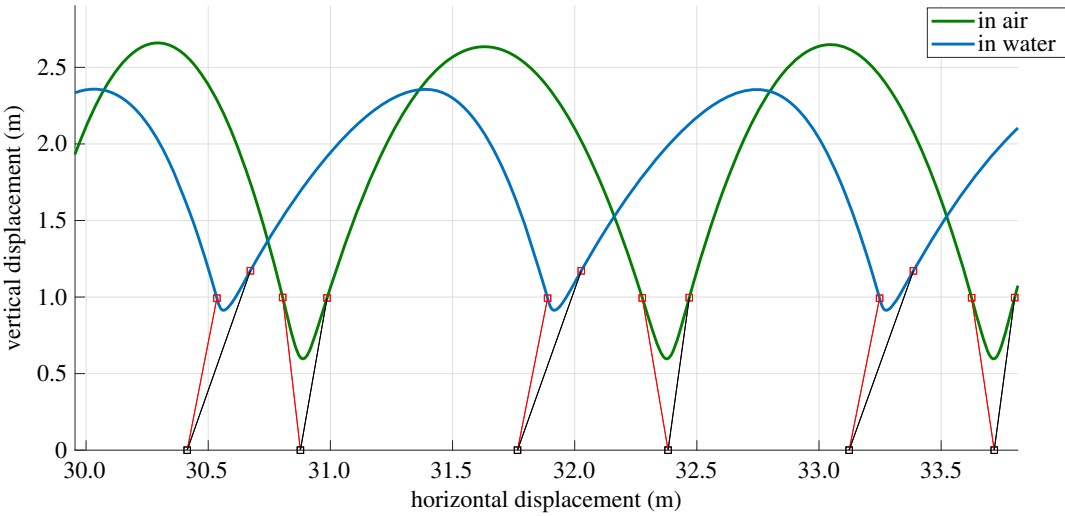

**Figure 13.** Gaits emergent from the universal controller in different mediums.

This emergent change in behaviour appears very similar to what is observed in a robotic salamander when the environment dictated the swap between walking and swimming gaits [14], and could explain the effectiveness of animals (including humans) to swiftly transition from land to low gravity or underwater environments [8,48]: the fundamental model is indeed the same, and by sensing the environment the locomotion is directly adapted. It is worth mentioning that, in our model, the change in density of the medium directly affects buoyancy and drag forces: both contributions could be plausibly detected by extero-perceptive or proprio-perceptive biological systems, such as the vestibular system.

It has to be noted, however, that prolonged exposure to low-gravity environment decreases locomotion capabilities during low-gravity to high-gravity transitions, e.g. on astronauts returning to the Earth, or going from space flights to the Moon [49]. These impairments are related to significant changes in skeletal, muscular and sensing systems, which alter the sensory-motor coordination [50]: nevertheless, the hypothesis of a unique locomotion model still holds while such defects could be linked to the impossibility of complying to the model. Eventually, in the very first Moon locomotion, astronauts mainly reported falls related to the interaction with the ground or structures (rocks, ladders, etc.), rather than on actual locomotion impairments [50], while the natural evidence of underwater pedestrian locomotion positively confirm the existence of a transferable skill.

Moreover, we claim that the abstract concept of balancing the CoM could be a reference strategy for legged locomotion in different media, and that legs' action is led by this balancing goal (i.e. learning to hop in place, or equivalently to reach $x_v = 0$). We report the learning progress of this strategy on different media (figure 3), and the performance in air is much better for the transferred agent (figure 7, middle column). With the help of the proposed cascaded controller, the model can reach a desired apex horizontal speed too, as shown in figure 9. The cascaded controller in water modifies the system to act like a highly damped spring system with small steady-state errors. Similarly, the controller transferred on land performs well in reaching the average target speed, with slight oscillations around the desired velocity (figure 10). The oscillations can be further reduced by adding a derivative term to the error values and the steady-state error can be reduced with an integral term.

Besides being no explicit references, to the authors knowledge, present in the biological literature, several lines of evidence supporting our 'learning to stop' approach. Termination of gait is considered to involve prediction of CoM position and speed [51], and how to place the foot accordingly; infants locomotion development involves the learning of braking actions [52]; facilitating actions in the real environment promotes an early skill acquisition [53]; and most bouts of young infants are made of a few steps only (one to three, before stopping without an apparent reason) [54]. These observations could promote a view of 'learning to stop', rather than explicitly 'learning to move'. This is also supported by the reward landscape for different objectives: learning to stop (figure 5) appears much smoother and continuous than learning to move at a constant velocity, figure 6. This enables training of gradient-based learning algorithms to be much more stable, robust and faster. Owing to the symmetric nature of the reward with respect to the initial conditions, the strategy is also well suited

for developing the cascaded architectures for building higher level controllers. Practically, such an approach will reduce the risk of damage as the controller tries to reduce the momentum of the system during the training process.

We do not presume that our hypothesis can explain all the facets of this complex research topic, however, it appears as a simple and, to the authors knowledge, plausible unifying objective to be taken into account during development. It is general enough to be resilient to change in body or environments (as happens continuously with children), it can be extended, with additional sensory input, to cluttered or complex surfaces (slippery grounds, gravel, grass, etc.), and it does not impose a hierarchical structure of skill developments, e.g. crawling as essential step for walking (which does not happen in all infants) [53]. Although we do not claim that the methodology presented here is actually employed by animals, we believe our results prove that a unique model could exist, and they envision learning to stop as a feasible control objective.

By comparing learning on the two different tested environments, we reported a clear asymmetry in the quality of the learned behaviour. Learning in water appears to be easy when compared with learning in air, and moreover the learnt control strategy could be positively transferred to the air environment (figure 3). Reward rapidly grew and settled on a high value for the training in water with respect to air, which supports the facilitating physical change proposed in [25]. Owing to the low inertial conditions and stabilizing effects of buoyancy in water, the reward landscape is much wider (figure 5), hence enabling faster and higher reward accumulation during training. With a massive number of training episodes ending at the first step for direct learning in air figure 4, the agent has reduced the number of chances to learn the correct control parameters: on the other hand, an agent transferred from the water environment allows an higher number of steps per training episode along with some prior knowledge of the control parameters, thus promoting the possibility of faster learning with fewer falls.

The last point we would like to discuss is related to the effectiveness of the transferred learning, i.e. water to land. The transferred agent performs surprisingly well both as overall reward (compared to air-only learning, figure 3) and as number of steps for training event (figure 4). The reward landscape explains the high performance of both in water and how it would facilitate the transferred agent (figure 5). With respect to previous works which explored the evolution of artificial creatures, it was already found how moving (swimming) in water was eventually beneficial for moving (crawling) on land [13]. Our results point in the same direction, pushing forward the same idea and promoting the concept that even legged agents can benefit from an initial learning phase in water. The use of fins as limbs for walking, jumping or crawling is reported in several fishes [55–57], and our results may suggest that effective land locomotion with limbs *must* be initiated in a water environment first, then transferred to shore, and eventually to land. The underwater environment was not only a convenient environment where legged animals evolved but it was the pivotal factor which enabled the development of limbed locomotion. The implication is that legged locomotion could not be produced directly on land, as it was too difficult and ineffective.

The presented results also has important implications on robotic learning approaches: the intrinsic self-stabilizing properties of the USLIP model allows the robotic hardware to experience less harmful impacts; diminishes the learning time and increases the learning episodes (pushes) for session. Eventually, the facility of learning in water and the possibility to transfer the learning on land may promote the adoption of on-line learning with the actual hardware, and to diminish the issues related to the reality gap between learning in simulations and in real environments.

## 5. Conclusion

Fundamental simplified models are powerful tools for studying and analysing complex dynamical systems. The SLIP and USLIP models have been fairly successful in explaining the observed gait patterns in nature with highly simplified mathematical models. In this work, we use these models along with the state of the art RL algorithms to study the transition of locomotion behaviour from water to air. Our findings show that transitioning from water to land is not just an evolutionary manifestation, but has significant advantages from a controller development perspective. We show that transferring knowledge from controllers and models learned in water is much more efficient and safer than developing them from scratch in air. This *shaping* strategy is tested in simulation using an RL algorithm called DDPG. Note that numerical optimization methods can be used to obtain control solutions for the SLIP and USLIP models. However, the obtained solutions are specific to the initial condition and the surrounding medium. DDPG allows us to develop an end-to-end controller

capturing the relationship between the controllers in various mediums. So, the proposed approach can be transferred to any robot with more complex dynamics. In order to unify and generalize the control strategy, we also propose the concept of 'learning to stop' as a novel locomotion objective that enables us to smoothly transition from one medium to the other while allowing us freedom to develop higher level controllers with a cascaded architecture. The underlying rationale behind the transfer learning principle and the performance of the controller are quantitatively analysed in this paper. This study not only provides corroborative evidence to the unification of locomotion models and behaviour in various environments but also proposes the idea of training robots in simpler environments as a efficient safer methodology for acquiring new skills. An extension of the work would be to investigate the ideal training conditions and scenarios for better transfer of skills in our environment to extraterrestrial environments.

Data accessibility. All the source code are available here: https://github.com/tomraven1/DDPG-hop. Data and relevant code for this research work are stored in GitHub: https://github.com/tomraven1/DDPG-hop and have been archived within the Zenodo repository: http://doi.org/10.5281/zenodo.4607469.

Authors' contributions. T.G.T. carried out the experiment. T.G.T. wrote the manuscript with support from G.P., M.C., F.I. and C.L. supervised the project.

Competing interests. We declare we have no competing interests.

Funding. This work was supported by Research England (Lincoln Agri-Robotics) as part of the Expanding Excellence in England (E3) Programme and by the SHERO project, a Future and Emerging Technologies (FET) programme of the European Commission (grant agreement ID 828818), Arbi Dario S.p.A. under the framework of Blue Resolution project, and by the National Geographic Society under the framework of GOLD (Guardian of the Oceans Legged Drone) project, grant no. NGS-56544 T-19.

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
