## [Peer Review File · Royal Society Open Science]

Review History

RSOS-210223.R0 (Original submission)

Review form: Reviewer 1

Is the manuscript scientifically sound in its present form?

Yes

Are the interpretations and conclusions justified by the results?

Yes

Is the language acceptable?

Yes

Do you have any ethical concerns with this paper?

No

Have you any concerns about statistical analyses in this paper?

No

Recommendation?

Accept as is

Comments to the Author(s)

The paper has been greatly improved and I think that it is ready for publication.

One final point is that I would still like to insist on running numerical optimization with SLIP and/or USLIP models to see how the performance of the controller obtained by DRL compares to the best possible one for a specific condition. This is not mandatory but I believe it would be interesting for readers because the comparison would give some intuition about tradeoff between generality and optimality.

Review form: Reviewer 2

Is the manuscript scientifically sound in its present form?

Yes

Are the interpretations and conclusions justified by the results?

Yes

Is the language acceptable?

Yes

Do you have any ethical concerns with this paper?

No

Have you any concerns about statistical analyses in this paper?

No

Recommendation?

Accept as is

Comments to the Author(s)

This revision is fine.

Decision letter (RSOS-210223.R0)

Dear Mr Picardi

On behalf of the Editors, we are pleased to inform you that your Manuscript RSOS-210223 "Learning to Stop: A Unifying Principle for Legged Locomotion in Varying Environments" has been accepted for publication in Royal Society Open Science subject to minor revision in accordance with the referees' reports. Please find the referees' comments along with any feedback from the Editors below my signature.

Please submit your revised manuscript and required files (see below) no later than 7 days from today's (ie 09-Mar-2021) date. Note: the ScholarOne system will 'lock' if submission of the revision is attempted 7 or more days after the deadline. If you do not think you will be able to meet this deadline please contact the editorial office immediately.

on behalf of Professor R. Kerry Rowe (Subject Editor)
openscience@royalsociety.org

Associate Editor Comments to Author:

In broad terms, the reviewers are satisfied that your paper is ready for publication, though a suggestion made by one of the reviewers appears sensible (and is recapitulated below). Incorporating their suggestion would add additional value to your work if it is possible to conduct this extra task:

"One final point is that I would still like to insist on running numerical optimization with SLIP and/or USLIP models to see how the performance of the controller obtained by DRL compares to the best possible one for a specific condition. This is not mandatory but I believe it would be interesting for readers because the comparison would give some intuition about tradeoff between generality and optimality. "

Reviewer comments to Author:
Reviewer: 1
Comments to the Author(s)

The paper has been greatly improved and I think that it is ready for publication.

One final point is that I would still like to insist on running numerical optimization with SLIP and/or USLIP models to see how the performance of the controller obtained by DRL compares to the best possible one for a specific condition. This is not mandatory but I believe it would be

interesting for readers because the comparison would give some intuition about tradeoff between generality and optimality.

Reviewer: 2

Comments to the Author(s)

This revision is fine.

===PREPARING YOUR MANUSCRIPT===

===PREPARING YOUR REVISION IN SCHOLARONE===

<https://royalsociety.org/journals/authors/author-guidelines/#supplementary-material> to include a suitable title and informative caption. An example of appropriate titling and captioning may be found at https://figshare.com/articles/Table_S2_from_Is_there_a_trade-off_between_peak_performance_and_performance_breadth_across_temperatures_for_aerobic_scops_in_teleost_fishes_/3843624.

Author's Response to Decision Letter for (RSOS-210223.R0)

See Appendix A.

Decision letter (RSOS-210223.R1)

Dear Mr Picardi,

I am pleased to inform you that your manuscript entitled "Learning to Stop: A Unifying Principle for Legged Locomotion in Varying Environments" is now accepted for publication in Royal Society Open Science.

on behalf of Professor R. Kerry Rowe (Subject Editor)
openscience@royalsociety.org

Appendix A

First of all, we would like to thank the Associate Editor and the Reviewers for their help and positive feedback throughout the revision process of our work.

The only point left to be discussed regards the comparison between our general control strategy obtained with reinforcement learning (RL) and an optimized strategy. We performed numerical optimization over the control space for one-step stopping for both U-SLIP and SLIP models and compared the control inputs obtained (angle of contact and elongation speed) with those obtained through the RL strategy in both underwater and terrestrial environments. In water the optimal action is obtained with lower elongation speeds as compared with the one derived from RL. This still holds in air, but with a smaller gap. Conversely, the optimal action in air exhibited higher angles of contact with respect to the RL one.

We included the results of this comparison through a paragraph (highlighted in red) at page 10 of the revised manuscript and adding a new figure to show the differences in control inputs (figure 8 of the revised manuscript).